# A large genomic insertion containing a duplicated follistatin gene is linked to the pea aphid male wing dimorphism

Binshuang Li[1†], Ryan D Bickel[1†], Benjamin J Parker[1‡], Omid Saleh Ziabari[1], Fangzhou Liu[1], Neetha Nanoth Vellichirammal[2], Jean-Christophe Simon[3], David L Stern[4], Jennifer A Brisson[1]*

[1]Department of Biology, University of Rochester, Rochester, United States; [2]University of Nebraska Medical Center, Omaha, United States; [3]INRAE, UMR 1349 IGEPP, Le Rheu, France; [4]Janelia Research Campus of the Howard Hughes Medical Institute, Ashburn, United States

**Abstract** Wing dimorphisms have long served as models for examining the ecological and evolutionary tradeoffs associated with alternative phenotypes. Here, we investigated the genetic cause of the pea aphid (*Acyrthosiphon pisum*) male wing dimorphism, wherein males exhibit one of two morphologies that differ in correlated traits that include the presence or absence of wings. We mapped this trait difference to a single genomic region and, using third generation, long-read sequencing, we identified a 120 kb insertion in the wingless allele. This insertion includes a duplicated *follistatin* gene, which is a strong candidate gene in the minimal mapped interval to cause the dimorphism. We found that both alleles were present prior to pea aphid biotype lineage diversification, we estimated that the insertion occurred millions of years ago, and we propose that both alleles have been maintained in the species, likely due to balancing selection.

*For correspondence:
jbrisso3@ur.rochester.edu

†These authors contributed equally to this work

Present address: ‡Department of Microbiology, University of Tennessee, Knoxville, United States

## Introduction

The evolutionary loss of flight ability in insects has long been leveraged to study the patterns and processes of adaptation (*Zera and Denno, 1997*; *Roff, 1990*; *Harrison, 1980*). The loss of flight has occurred repeatedly across insect orders, such as ants, termites, beetles, crickets, and aphids (*Zera and Denno, 1997*), with such evolutionary shifts to flightlessness co-occurring with entry into stable or isolated habitats where flight is unnecessary (*Roff, 1990*; *Wagner and Liebherr, 1992*). Loss of flight allows animals to shift resources toward production of offspring instead of flight appendages and fuel. Flight-capable and flight-incapable insects therefore present an evolutionary tradeoff between dispersal and reproduction (*Zera and Denno, 1997*; *Harrison, 1980*). This tradeoff is sometimes observed within species, where some individuals develop with wings and others are wingless. We focus here on wing dimorphisms that are under genetic control (*Roff, 1986*), although some species display phenotypically plastic wing dimorphisms (*Zera and Denno, 1997*).

Wing dimorphisms offer an appealing system for exploring the genetic basis of adaptation because of their relatively simple genetic architectures. In species of weevils, ground beetles, and dung flies, for example, a single locus controls each species' wing dimorphism (*Roff, 1986*). Other dimorphisms are polygenic, but *Roff (1986)* found that eight of 22 species exhibited Mendelian inheritance. Transitions between flight-capable and flight-incapable morphs have probably evolved thousands of times in insects (*Wagner and Liebherr, 1992*; *Whiting et al., 2003*). Despite the importance of alternative winged and wingless morphs in insect adaptation, none of the loci controlling wing dimorphism have yet been characterized (*Saastamoinen et al., 2018*).

Here, we investigated the genetic basis and evolution of the pea aphid (*Acyrthosiphon pisum*) male wing dimorphism. Pea aphid males exhibit distinct winged and wingless adult phenotypes (*Figure 1A*) and differ in a set of correlated traits. Winged males have well-developed fore and hind-wings, and direct and indirect flight muscles, while wingless males have a smaller thorax, and lack wings and associated flight musculature (*Dixon and Howard, 1986*; *Braendle et al., 2006*; *Ogawa et al., 2012*). The males also differ behaviorally; winged males are more active than wingless males and thus may gain more mating opportunities with females in competitive trials (*Sack and Stern, 2007*), while the wingless males do not fly and thus avoid the high mortality associated with flight. Unlike the aphid asexual female polyphenism where environmental cues like high density and low food quality lead to the production of winged versus wingless female morphs (*Dixon, 1973*), the male polymorphism is genetically determined. A single locus with two alleles on the X chromosome called *aphicarus* ('aphid' plus 'Icarus', *api*) controls male morphology (*Braendle et al., 2005*; *Caillaud et al., 2002*). Male aphids carry only a single X chromosome (females are XX, males are XO), so males carry only a single winged or wingless allele. The correlated anatomical and behavioral traits suggest that the causal locus can regulate development, and perhaps physiology, in multiple anatomical domains. We found that a large structural variant, resulting from a duplication event, underlies the winged versus wingless differences. We then explore the evolutionary history of this locus, including the maintenance of the duplication polymorphism and its distribution in present-day populations.

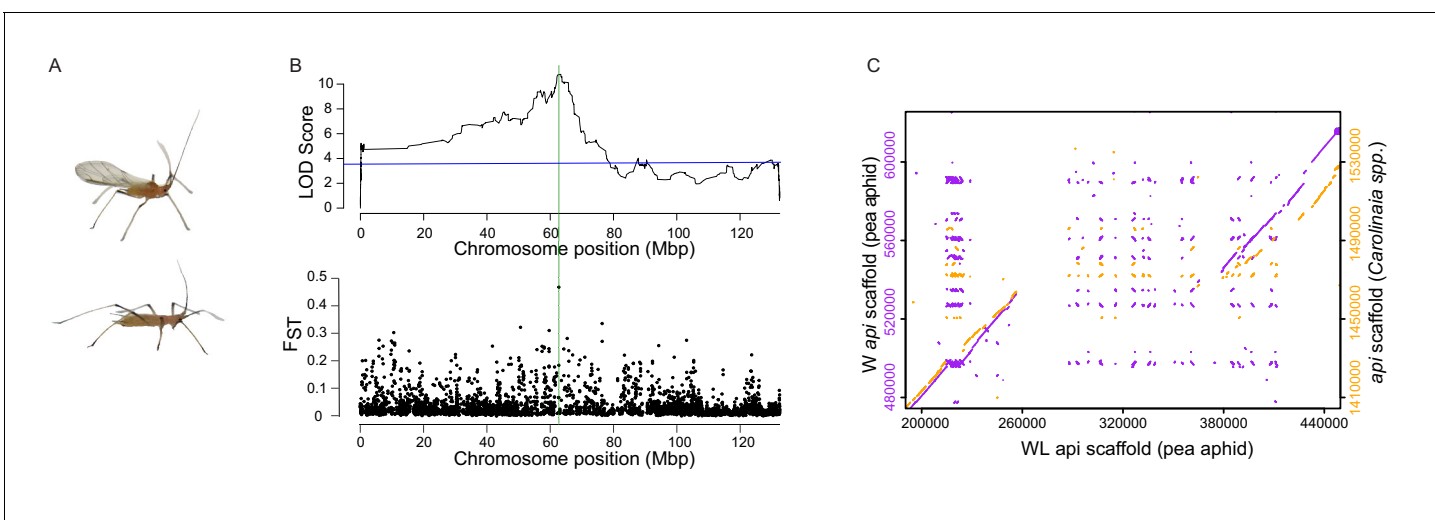

**Figure 1.** Linkage and association mapping of *api*. (A) Winged (top) and wingless (bottom) males. (B) Top: QTL analysis revealed the presence of a single QTL peak on the X chromosome as assembled by *Li et al. (2019)*. The blue line indicates p=0.01 determined by 1000 permutations of the phenotype data relative to the genotype data. Further RFLP analysis of recombinant F2 individuals localized *api* to a ~ 190 kb region, which is highlighted by a green, vertical line. Bottom: Genetic differentiation between winged and wingless males using $F_{ST}$ values calculated from 20 kb windows across the X chromosome. (C) Dot plots showing areas of similarity among our newly assembled pea aphid genomic scaffolds containing *api* (either W:winged or WL:wingless, as indicated) and the scaffold containing *api* from a species in the *Carolinaia* genus.

The online version of this article includes the following figure supplement(s) for figure 1:

**Figure supplement 1.** Wingless, but not winged reads, map to the wingless-specific region.

**Figure supplement 2.** Similarity of the wingless-specific region to putatively homologous regions in the peach-potato aphid (*Myzus persicae)*, the *Carolinaia spp.* aphid, and the Russian wheat aphid (*Diuraphis noxia*).

**Figure supplement 3.** The male wing-dimorphic pea aphid (*Acyrthosiphon pisum*, arrow) is embedded in a clade of mainly monomorphic winged species.

**Figure supplement 4.** Similarity of the wingless-specific region to an autosomal scaffold containing the *fs-2* gene.

## Results and discussion

### Mapping the *api* locus

The *api* locus is located on the X chromosome, which is estimated to contain a third of the pea aphid genome (*Jaquiéry et al., 2018*). The locus was previously localized to a 10 cM region (*Braendle et al., 2005*), a large region consisting of hundreds of genes. We took a two-pronged mapping approach to narrow down the *api* region. First, we crossed males and females from the F1 line from the original mapping population (*Braendle et al., 2005*) to generate additional F2 individuals. We performed multiplexed shotgun genotyping (*Andolfatto et al., 2011*) to simultaneously identify and score single nucleotide polymorphisms (SNPs) from 164 of these F2s. QTL analysis of these data using a recently published chromosome-level assembly (*Li et al., 2019*) resulted in a single LOD peak on the X chromosome (LOD 10.8, with a 1% LOD permutation threshold of 3.6, *Figure 1B*, top; *Source data 1*). Second, we performed genome-wide association mapping using 44 winged and 44 wingless males collected across the U.S. from alfalfa (*Medicago sativa*) fields (*Supplementary file 1*). We pooled the 44 winged and wingless males separately and performed Illumina resequencing to high coverage for a total of 68X and 70X coverage, respectively. $F_{ST}$ analysis between the winged and wingless sequenced pools revealed a single, large $F_{ST}$ value (*Figure 1B*, bottom; *Source data 2*) close to the LOD peak from the QTL analysis.

We further refined the *api* region by assaying restriction fragment length polymorphism (RFLP) markers on a panel of 40 F2 individuals that each carried a recombination event between the previous closest *api* flanking markers (*Braendle et al., 2005*). Two markers displayed perfect association with all 40 F2s, narrowing the *api* interval to a ~ 190 kb region (see the green vertical line in *Figure 1B*, spanning positions from 62.56M to 62.75M). Hereafter, we will use the term '*api* region' to refer to this genetically defined region.

We then used an additional set of fully sequenced pea aphid genomes to independently verify the genomic location of *api*. The pea aphid is a species complex with as many as 15 host specialized lineages, called biotypes, that probably began diverging 500,000 years ago (*Fazalova and Nevado, 2019*, but see *Peccoud et al., 2009b*). Biotypes have limited gene flow between them, and the ones with the least genetic exchange have been described as incipient species (*Peccoud et al., 2009a*; *Peccoud et al., 2009b*). We used the complete genomes of 23 individual genotypes from nine different biotypes to investigate patterns of association in the *api* region: nine winged allele carrying genotypes from five biotypes, and 14 wingless allele carrying genotypes from six biotypes (two biotypes, pea and alfalfa, had individuals of both allele classes; we sequenced males of these individuals to separately sequence the two X chromosomes). SNPs across the *api* region segregated perfectly with the male winged or wingless phenotype across these biotype samples. Thus, data from linkage mapping, and the association study within and between biotypes, all confirm the same genomic location of the causative *api* locus.

### Genome sequencing reveals a structural variant that differentiates the two alleles

Given the lack of a contiguous reference sequence in the *api* region in the previous assembly of the reference genome (v2.0), we separately sequenced the genomes of an F2 female homozygous for the winged (W) allele to 28x coverage and an F2 female homozygous for the wingless (WL) allele to 11x coverage using Nanopore long-read sequencing. For each allele, we assembled contiguous sequences that spanned the entire *api* region (WL scaffold ID = utg000342l; W scaffold ID = tig00159082l). We identified a ~ 120 kb insertion in the WL scaffold relative to the W scaffold (*Figure 1C*, left axis). The SNPs that originally identified this genomic region in the linkage and association mapping are adjacent to, and thus linked to, this insertion. The pool-seq reads from wingless males (generated for the association mapping discussed above) generated the same level of coverage across this insertion (including reads that cross the insertion boundaries) as the surrounding sequence, while the winged male pool-seq reads did not map to this inserted region, confirming the absence of this sequence in winged male genomes (*Figure 1—figure supplement 1*). This 120 kb insertion is not assembled correctly in the original pea aphid genome published in *International Aphid Genomics Consortium (2010)*, which is homozygous for the wingless allele, so a portion of this region is present in that assembly as a separate contig. In the more recent, near-

chromosome length assembly of the pea aphid genome (*Li et al., 2019*), the authors reported linkage of an approximately 50 kb insertion to the *api* locus. However, we found that the *api* region is incompletely assembled in this new assembly as well. We therefore use our new, wingless assembly for the remainder of this work.

To determine whether this indel represents an insertion in the wingless allele or a deletion in the winged allele, we aligned the winged and wingless *api* scaffolds against homologous genomic scaffolds from three other aphid species, the peach-potato aphid (*Myzus persicae*), an unidentified species from the *Carolinaia* genus (*Carolinaia spp.*), and the Russian wheat aphid (*Diuraphis noxia*) (other available aphid genomes were not used because they did not have scaffolds that spanned this region). The peach-potato aphid produces only winged males and the Russian wheat aphid produces only wingless males. We do not know the male phenotype of the *Carolinaia* species. We observed that the pea aphid wingless scaffold contained ~120 kb of sequence not present in other species, while the pea aphid winged scaffold exhibited nearly complete synteny across the region (*Figure 1C*, right axis, for comparison to the *Carolinaia spp.*; see *Figure 1—figure supplement 2* for comparison to all three aphid species). Thus, the ~120 kb sequence in the wingless allele is an evolutionarily derived insertion that occurred in the lineage that led to pea aphids. The majority of the species closely related to the pea aphid produce only winged males (*Figure 1—figure supplement 3*), so the wingless phenotype is likely the derived phenotype. Thus, both the wingless male phenotype and the insertion appear to be derived characteristics of pea aphids.

To investigate the source of this inserted sequence, we used a BLASTn analysis of the ~120 kb wingless-specific region against our newly reconstructed pea aphid genome. This analysis revealed that approximately half of the inserted sequence showed similarity to another region of the reference genome (scaffold utg000238l; *Figure 1—figure supplement 4*), which is on an autosome. Thus, most of the inserted sequence at *api* is derived from a duplication of an autosomal region. The remainder of the insertion is largely composed of complex repeats which are likely the result of mobile element insertions that occurred after the duplication event.

## A candidate *follistatin* gene in the inserted sequence

Annotation software (Augustus v.3.3.1) predicted 12 open reading frames (ORFs) in the wingless *api* insertion. The gene models located within the insertion in *Figure 2a* are from this annotation; models outside the insertion are the genome v3.0 annotations (*Li et al., 2019*). 11 of the 12 ORFs (*Supplementary file 1*) appear to be remnants of transposable element insertions. Ten show nucleotide similarity to previously identified pea aphid transposable element sequences (*International Aphid Genomics Consortium, 2010*). Another, ORF g6, has a DDE_Tnp4 superfamily domain (7.15e-3), which is part of the DDE superfamily of endonucleases that are likely transposases. Its predicted protein sequence is highly repetitive, with over 100 copies present across the pea aphid genome (tBLASTn evalue < −20). 38% of the pea aphid genome has been identified as derived from transposable element sequences (*International Aphid Genomics Consortium, 2010*), thus the *api* insertion is not unique in containing these elements. The remaining ORF, *fs-3*, is not repetitive and, importantly, is the only ORF that has homologous sequence in the insertion's autosomal source location.

The ORF that originated with the insertion shows high similarity to the *Drosophila melanogaster follistatin* (*fs*) gene (blastp e-value of $4e^{-43}$). In *Drosophila*, *fs* is a secreted protein that modulates TGF-β signaling by interacting antagonistically with activin (*Bickel et al., 2008*; *Pentek et al., 2009*). The *Drosophila* Fs protein contains follistatin-Kazal domains necessary for activin binding (*Wang et al., 2000*; *Keutmann et al., 2004*). Three follistatin-Kazal domains are present in the pea aphid *fs* gene that is in the wingless-specific insertion.

We searched the pea aphid genome for *fs* homologs and identified three total *fs* homologs: the *fs* homolog in the polymorphic insertion (*fs-3*), the *fs* homolog in the source of the insertion (*fs-2*), and a third homolog (*fs-1*). A search of four other published aphid genomes revealed only a single *fs* copy in each species. The other aphid genomes are significantly diverged from the pea aphid; the most closely related genome to the pea aphid is that of the peach-potato aphid, which diverged from the pea aphid over 40 million years ago (*Kim et al., 2011*). We therefore used the Illumina technology to sequence the genome of the rose aphid, *Macrosiphum rosae*, which diverged from the pea aphid more recently (*Figure 2B*; *Kim et al., 2011*; *Hardy et al., 2015*), although the exact divergence time is unknown. We examined the relationships among the *fs* homologs relative to the

species tree (*Figure 2B,C*). *fs-1* has a slower rate of evolutionary change compared to the *fs-2/fs-3* lineage, suggesting that *fs-1* likely retains the ancestral *follistatin* function.

The *fs* gene tree (*Figure 2C*) shows that there have been two *fs* duplication events. The first duplication created *fs-1* and *fs-2/3*. It occurred around the time that the Macrosiphini aphids represented here (the peach-potato, *Carolinaia spp.*, rose, and pea aphid) diverged from one another about 42 mya (*Kim et al., 2011*). The rate of synonymous substitution (dS) between *fs-1* and the *fs-2/fs-3* lineage is consistent with this divergence time: the average dS between the peach-potato aphid and the pea aphid is 0.25 (*Ji et al., 2016*), while dS between *fs-1* and *fs-2* or *fs-3* are very similar at 0.28 and 0.29, respectively.

Despite this ancient duplication, of the species examined here, we found *fs-2* and *fs-3* only in the pea aphid genome (and not the rose aphid or the *Carolinaia spp.* aphid). More recently, *fs-2/3* duplicated to produce *fs-2* (the autosomal, *api*-source copy) and *fs-3* (the copy found solely in the pea aphid wingless allele). The most parsimonious explanation for these patterns is that the initial duplication that gave rise to the *fs-2/3* lineage arose after the split of the pea aphid and the peach-potato aphid, and this duplicate copy was subsequently lost in the rose aphid lineage. The ambiguity of the timing, and the limited taxon sampling, however, do not preclude other evolutionary scenarios. Future sequencing of more aphid genomes would provide insight into the timing of the emergence of these paralogs.

All pairwise dN/dS comparisons using *fs* copies in *Figure 2C* resulted in values less than one (range of 0.03–0.21, *Supplementary file 2*), indicating purifying selection acting on all copies. Thus, all three *fs* paralogs seem to be functional genes. Although all three *fs* genes appear to be functional, *fs-3* is only present in the WL insertion, which is not found in winged males. Therefore, *fs-3* is not required for aphid survival, although it appears to provide a benefit to individuals that carry it.

To examine if *fs-3* is expressed in wingless males, we performed RT-PCR on cDNA collected from winged and wingless males across three developmental stages (late stage embryos, 1st instar nymphs, and 2nd instar nymphs, *Figure 2D*). We did not perform RT-qPCR for this gene because our experimental design was aimed at confirming that the gene was indeed expressed, and not to perform an analysis of comparative expression levels. The winged allele lacks *fs-3* and thus winged males show no expression (*Figure 2D*). In contrast, the wingless males expressed *fs-3* (*Figure 2D*). Female samples are also shown for comparison. Winged and wingless males are morphologically distinct by the second nymphal instar (*Ogawa et al., 2012*), so the developmental timepoint of morph determination must occur before this stage. In the phenotypically plastic pea aphid females, wing morph determination occurs embryonically (*Sutherland, 1969*). We reasoned that the action of *api*, and thus morph determination, likely begins at the embryonic stage and possibly persists throughout development. *fs-3* gene expression present at these early developmental stages is consistent with this expectation. *fs-3* is a strong candidate gene for causing the wingless phenotype, but other genes within or flanking the WL insertion cannot be ruled out as contributing to, or causing, male winglessness. For example, there is high sequence divergence between the winged and wingless alleles in the flanking sequence. In the 10 kb to the right of the insertion we observe ~6% divergence in the alignable portion of the sequence, plus indels that range from 1 to 920 bp. Many SNPs in the flanking region are strongly associated with the male phenotype (*Figure 2—figure supplement 1*).

We examined whether gene expression patterns for other genes in this region are consistent with their potential role in causing male winglessness. There are five genes present in the pea aphid genome annotation that are outside of the insertion, but inside or partially inside the minimal *api* region (pea aphid genome annotation v3.0, *Figure 2A*; *Supplementary file 1*). We provided names to these five genes as follows: two aphid-specific genes (*as1*, *as2*; neither gene contains conserved domains and therefore their possible functions are unknown), a predicted mitochondrial sorting and assembly gene (*sam50*), a predicted fibroblast growth factor receptor substrate gene (*frs2*), and a predicted chromodomain-encoding gene (*cdg*). We found no nonsynonymous polymorphisms in these genes that perfectly segregate with the *api* phenotype across the pool-seq and biotype data. Any mutations that contribute to the winged and wingless males would, therefore, likely be regulatory. We therefore measured the expression levels of the five genes (*as1*, *as2*, *sam50*, *frs2*, and *cdg*) using RT-qPCR. We focused on two developmental stages, embryos and first instar nymphs, as the likely critical times as explained above. *as2* is not expressed in males at these stages. Of the four other genes (*as1*, *sam50*, *frs2*, and *cdg*), none were significantly differentially expressed after multiple comparison correction (*Figure 2—figure supplement 2*, although *as1* embryos had an

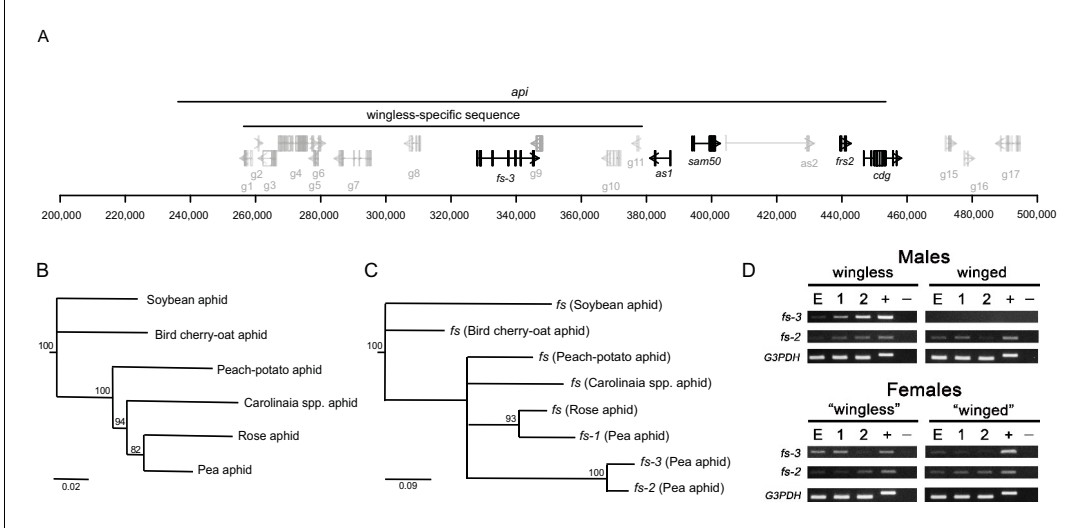

**Figure 2.** An expressed *follistatin* gene duplicate in the wingless-specific insertion. (**A**) Gene models at the *api* locus, with distances shown in bp on the x-axis. The recombination mapping defined *api* region and the wingless-allele specific insertion are indicated. ORFs inside the wingless-specific insertion are our own annotation, while those outside the insertion are those of the genome v3.0 (*Li et al., 2019*). Repetitive ORFs and those outside of the *api* region are shown in gray. ORFs with evidence for expression either from RT-PCR or RT-qPCR are shown in black. (**B**) Phylogenetic tree of the aphids studied here: the soybean aphid (*Aphis glycines*), the bird cherry-oat aphid (*Rhopalosiphum padi*), the peach-potato aphid (*Myzus persicae*), an unidentified *Carolinaia* genus species, the rose aphid (*Macrosiphum rosae*), and the pea aphid (*Acyrthosiphon pisum*). The maximum likelihood tree was constructed from 2,378 bp of DNA sequence from elongation factor 1α, 18S rRNA, 12S rRNA, cytochrome b, and cytochrome c oxidase subunit I collected from Genbank. (**C**) Maximum likelihood tree of the different copies of *follistatin* (*fs*) nucleotide sequences. Nodes with bootstrap values < 80 are collapsed. (**D**) RT-PCR analysis of *fs-3* and *fs-2* gene expression across different developmental stages (E: embryo cDNA, 1: 1st instar nymph cDNA, 2: 2nd instar nymph cDNA; +: genomic DNA, -: no cDNA) of wingless males, winged males, daughters of crowded adult wingless females ('winged' females, which are predominantly winged females), daughters of uncrowded adult wingless females ('wingless' females, which are predominantly wingless female samples). Females here are heterozygous for the *api* locus, and thus have *fs-3* as indicated by the positive band in the female gDNA lanes. Data for the G3PDH gene are provided as a positive control.

The online version of this article includes the following figure supplement(s) for figure 2:

**Figure supplement 1.** Association between SNPs at *api* and the male wing phenotype.

**Figure supplement 2.** Genes in the insertion-adjacent region are not differentially expressed as measured by RT-qPCR analysis.

uncorrected t-test p-value=0.013; *Source data 3*). We therefore found no strong evidence that these genes contribute to the male wing dimorphism.

Multiple lines of evidence suggest that *fs-3* is the male morph determination gene. *fs-3* is present in all wingless males and absent from winged males. It is not essential for aphid survival because winged males do not have it. Yet, *fs-3* is under purifying selection and its coding sequence has persisted, intact, for thousands of generations. The maintenance of *fs-3* suggests that it provides a selective advantage, but this advantage can be present only in individuals that have *fs-3* (*i.e.*, only wingless males or females carrying this allele). Furthermore, *fs-3* is expressed at the relevant time for mediating morph determination. A transgene rescue or mutational knock-out experiment (ideally both) are ultimately required to determine whether *fs-3* causes the male wing polymorphism.

The function of *fs*, as studied in *Drosophila*, provides insight into how *fs-3* might function in the pea aphid male wing polymorphism. The sole *Drosophila fs* gene encodes a protein, Fs, that binds activins to affect TGF-β/activin signaling (*Bickel et al., 2008*; *Pentek et al., 2009*). Overexpression studies (*Bickel et al., 2008*; *Pentek et al., 2009*) revealed two potentially relevant Fs roles: regulation of cell proliferation and regulation of ecdysone receptor expression in the brain. For the former, wings were smaller in *fs* overexpression mutants (*Bickel et al., 2008*; *Pentek et al., 2009*). In the pea aphid wingless males, expression of an additional *fs* copy could negatively regulate cell proliferation in the wing and wing musculature, resulting in a lack of growth. For the latter, *fs* overexpression caused phenotypes that mimicked disruption of ecdysone signaling; the authors posited that there is a potentially underappreciated interaction between activin signaling and ecdysone responses (*Pentek et al., 2009*). This is particularly intriguing because the pea aphid plastic, female

wing dimorphism is regulated, at least in part, by ecdysone signaling (*Vellichirammal et al., 2017*). Pea aphid *fs-3* might, therefore, directly or indirectly activate ecdysone signaling in a tissue-specific manner during the development of the wingless morph, causing it to display the wingless male phenotype. So, rather than using an environmental cue to activate ecdysone signaling and to ultimately cause winged versus wingless developmental events, as in the pea aphid female polyphenism, the males may begin the process using the *fs-3* gene product. Future work is required to determine whether or not *fs-3* influences TGF-β/activin signaling in the pea aphid and how these changes might alter the male phenotype.

## Evolution at the *api* locus

To investigate the evolution of the *api* region within the pea aphid species complex, we used the 23 individually sequenced pea aphid genomes from nine biotypes discussed above. Of these genomes, all 14 wingless individuals carried the insertion (*Figure 3—figure supplement 1*) and the nine winged individuals did not (*Figure 3—figure supplement 2*). Phylogenetic network analysis using a region of the aphid genome outside of *api* (*Source data 4*) showed that genetically distinct biotypes, such as those feeding on *Lathyrus pratensis* (Lap1-3) and *Vicia cracca* (Vc1-2), group together and are separated by relatively long branches (*Figure 3A*, top). The other biotypes exhibited little network structure, suggesting that there is gene flow among these biotypes, as reported previously (*Peccoud et al., 2009a*). In stark contrast, within the *api* region but just outside of the insertion (*Source data 5*), we found that the winged and wingless alleles are separated by a long branch, implying that biotype differentiation occurred after the origin of the WL *api* allele (*Figure 3A*, bottom). Thus, the WL *api* insertion must predate the biotype divergence that occurred ~500,000 ya.

To further explore the age of the insertion, we used the synonymous divergence rate, dS, as a measure of neutral differences and thus as a proxy for age. We first explored the distribution of dS values for all genes between two divergent biotypes, a *Lathyrus pratensis* biotype genome (Lap1 from *Figure 3A*) and a *Medicago sativa* biotype genome (Ms3 from *Figure 3A*). Even between these

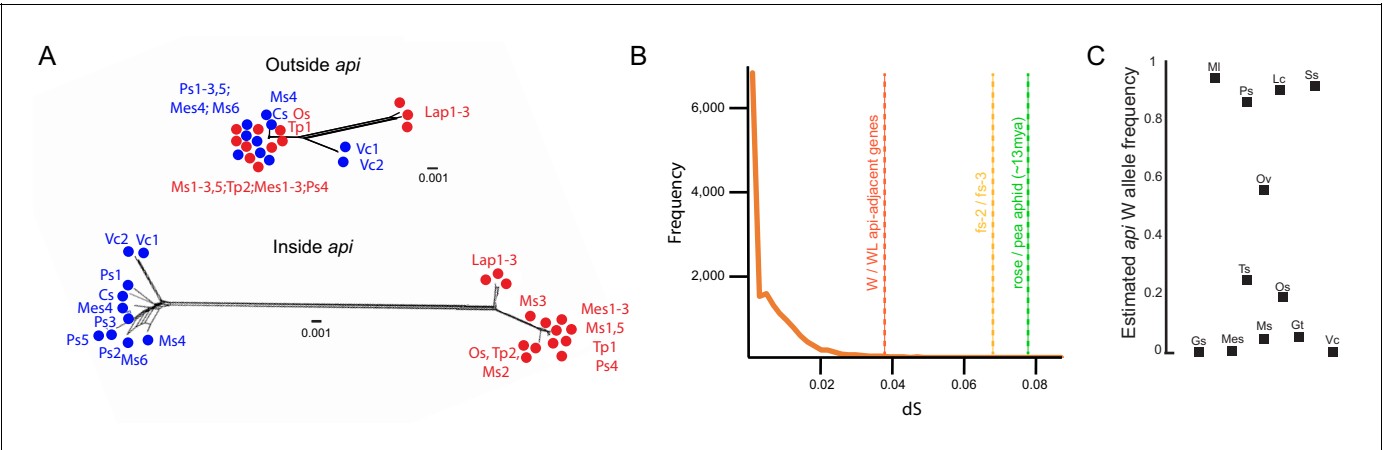

**Figure 3.** The polymorphism predates radiation within the species complex. (**A**) Tree networks, built with SplitsTrees, based on positions ~ 1,235 kb-1,245kb ('Outside *api*') or ~379 kb–388 kb ('Inside *api*') on the WL *api* scaffold. Winged individuals are colored in blue and wingless alleles in red. Abbreviations: Cs: *Cytisus scoparius*, Lap: *Lathyrus pratensis*, Mes: *Melilotus spp.*, Ms: *Medicago sativa*, Os: *Ononis spinosa*, Ps: *Pisum* sativum, Tp: *Trifolium pratense*, and Vc: *Vicia cracca*. (**B**) Neutral divergence times, as measured by dS. Orange line shows the dS distribution for 16,367 genes, derived from comparing individuals Lap1 and Ms3. Dotted vertical lines indicate the dS values for the average of the three *api*-adjacent genes (*as1*, *sam50*, *frs2*) when winged and wingless individuals are compared, the comparison of *fs-2* and *fs-3*, and the average rose to pea aphid dS. (**C**) Allele frequency estimates based on SNP frequency analysis of read counts in pool-seq data for each of the different pea aphid biotypes. Biotypes that overlap with data in (**A**) are shown with the same abbreviations as in (**A**), but some biotypes are unique to analyses in (**A**) or (**C**). Additional abbreviations: Gs: *Genista sagittalis*, Gt: *Genista tinctoria*, Lc: *Lotus corniculatus*, Ml: *Medicago lupulina*, Ov: *Onobrychis viciifolia*, Ss: *Securigera spp.*, and Ts: *Trifolium spp.*

The online version of this article includes the following figure supplement(s) for figure 3:

**Figure supplement 1.** Reads from wingless individuals of different biotypes map to the wingless-specific region.

**Figure supplement 2.** Reads from winged individuals of different biotypes do not map to the wingless-specific region.

divergent biotypes, most genes had few differences, with a median dS of 0.0035 (orange distribution, *Figure 3B*; *Source data 6*). In contrast, the three expressed genes with exons within the *api*-adjacent region had dS values between winged and wingless alleles that are an order of magnitude higher than the average of all genes: 0.045 (as1), 0.035 (sam50), and 0.036 (frs2) (vertical dashed line in *Figure 3B*). Even more strikingly, dS between *fs-3*, the *follistatin* copy in the WL insertion, and *fs-2*, the *follistatin* copy at the autosomal origin, was 0.068 (*Figure 3B*, vertical dashed line). These high dS values also support the hypothesis that the *api* WL insertion is considerably older than the biotype diversification.

We next estimated the age of the insertion relative to the divergence between the pea aphid and the rose aphid (*Macrosiphum rosae*). The pea aphid and rose aphid display an average dS of 0.08 (*Figure 3B*). Assuming a molecular clock, and an estimated dS and divergence time between the pea aphid and the peach-potato aphid (*Myzus persicae*) of 0.25 and approximately 42 million years, respectively (*Ji et al., 2016*), we estimate that the pea aphid and rose aphid diverged approximately 13 million years ago. We therefore conclude that the *api* WL insertion occurred approximately 10 millions years ago. Given the age of this insertion, the region is likely not unique to pea aphids, but rather could be responsible for the wingless male phenotype observed in other, related species as well. For example, a species not examined in the phylogeny shown in *Figure 1—figure supplement 3* is *Acyrthosiphon auctum*, which is purportedly within the same genus as the pea aphid, but this species only produces wingless males (*Blackman and Eastop, 2000*). It will be interesting to interrogate other, closely related wingless and dimorphic species to determine whether or not the insertion is present in them and thus responsible for the wingless phenotype in other species.

To explore contemporary frequencies of the two alleles, we estimated the winged allele frequency from 12 different pea aphid biotypes. To do this, we focused on a noncoding SNP in the *api* region that segregates perfectly with the male wing phenotype across all 23 sequenced genomes from the nine biotypes, and which is 95% associated with the male phenotype in the alfalfa-biotype-specific, pool-seq data. We examined this SNP from pre-existing pool-seq data collected from each of the 12 biotypes (*Guyomar et al., 2018*) to estimate the winged allele frequencies. Each pool contained from 14 to 35 individuals from France, which has high host plant diversity, and each pooled individual had a distinct, multilocus genotype (*Guyomar et al., 2018*). The majority (9/12) of the biotype lineages exhibited both alleles, with the ancestral, winged allele frequency ranging from 5% to 94% (*Figure 3C*). In the remaining three biotypes (Gs, Mes, and Vc), the derived, wingless allele has gone to fixation in the samples of our dataset. This inter-biotype variation raises the intriguing idea that specialization on distinct legume hosts may influence male wing dimorphism variation.

The winged and wingless alleles have been maintained within the pea aphid lineage for millions of years. This variation may have been maintained within one or more biotypes as we see in modern-day sampling, or alternatively, the pea aphid species *complex* has maintained both alleles for thousands of years, but the alleles have moved between individual biotype via introgression. Unlike the female pea aphid wing dimorphism, which is well studied in terms of the investment tradeoff in reproduction versus dispersal (*Zera and Denno, 1997*), it is not as clear what fitness advantage a winged or wingless pea aphid male has over its counterpart. Indeed, the existence of wing dimorphisms in male insects are less well understood generally, especially given their limited energetic investment in offspring relative to females (reviewed in *Langellotto et al., 2000*). Hypotheses would include inbreeding avoidance and reduction of local mate competition driving the increase in abundance of winged males, while the high mortality associated with flight would drive an increase in wingless males. Future studies should, therefore, aim to understand why both morphs have existed for so long in pea aphids.

## Conclusions

We have identified a derived insertion carrying a duplicated *follistatin* homolog that underlies the differences between winged and wingless males. This study represents an important breakthrough in understanding the molecular evolution of wing dimorphisms, which have evolved repeatedly in insects (*Roff, 1986*). Furthermore, this work demonstrates that a dramatic phenotypic difference is due to a relatively old insertion containing a gene duplicate, and thus that mutations of relatively large effect can play an important role in adaptive evolution (*Orr and Coyne, 1992*). The discovery of these types of structural variants underlying morphological adaptations have largely gone undetected (at least, compared to SNPs, although notable examples exist e.g., *Lamichhaney et al.,*

*2016*; *Joron et al., 2006*) to date due to technological limitations. Long-read genome sequencing technologies are likely to uncover additional cases of structural variants underlying morphological evolution in the future.

## Materials and methods

### Linkage mapping

*Braendle et al. (2005)* previously established an *api* linkage mapping population. We used the F1 line from that population to generate additional F2 recombinants. Specifically, F1 asexual females (that reproduce through parthenogenesis) were placed on *Vicia faba* plants in an incubator at 16°C and a photoperiod of 12 hr light and 12 hr dark. After two generations of asexual reproduction, these conditions induce the production of sexual females and males. F1 males are both winged and wingless. Females produce males asexually by losing one X chromosome (*Blackman, 1987*). The *api* locus is on the X chromosome, so F1 females produce either winged or wingless males depending on which chromosome they drop. Male siblings are therefore identical on their autosomes, but they carry different X chromosomes. We crossed F1 females to F1 males, collected fertilized eggs, sterilized them in 1% calcium propionate on Whatman paper, and placed them in an incubator that alternated between 4°C for 12 hr light and 0°C for 12 hr dark. After 90 days, eggs were removed from this incubator and placed in a 19°C incubator that alternated between 16 hr light and 8 hr dark. F2 hatchlings were transferred to individual plants for asexual reproduction to establish a line of that F2 individual.

Genomic DNA from 164 F2 females was isolated, quantified, and diluted to 2 ng/µl and subjected to multiplexed shotgun genotyping (*Andolfatto et al., 2011*). The final amplified libraries were sequenced on an Illumina Genome Analyzer. We generated parental X-chromosome genomes for the MSG software pipeline (https://github.com/YourePrettyGood/msg; copy archived at https://github.com/elifesciences-publications/msg; *Pinero et al., 2020*) by updating a recently published assembly of the pea aphid genome (*Li et al., 2019*) with short sequence reads generated from wingless and winged males from the F1 strain and a custom script (available upon request). MSG produced genotype posterior probabilities that were imported directly into the genotype data structure of R-qtl (*Broman et al., 2003*) using a custom script (https://github.com/dstern/read_cross_msg; copy archived at https://github.com/elifesciences-publications/read_cross_msg; *Stern, 2020*). We performed QTL analysis assuming a single additive locus for winged versus wingless males using the R-qtl command scanone using Haley-Knott regression (*Haley and Knott, 1992*). Genome-wide significance values were calculated using 1000 permutations of the phenotype data over the genotypes.

### Pool-seq

We used 44 winged and 44 wingless male pea aphids induced from females collected from Nebraska, New York, California and Massachusetts (*Supplementary file 3*). Genomic DNA (gDNA) from each male was extracted using the Qiagen DNeasy Blood and Tissue Kit. 50 ng of gDNA from each male was pooled together with males of the same phenotype. Paired-end libraries were prepared with the TruSeq DNA PCR-Free Library Preparation Kit at the University of Rochester Genomics Research Center and sequenced on an Illumina HiSeq2500 Sequencer with paired 125nt reads. First, reads were mapped to the recently published whole-X-chromosome assembly (*Li et al., 2019*) using bowtie2 v2.2.9 (50) with default parameters. The SAM files were converted to BAM files using samtools (v1.7) and later used for Fst analysis. In addition, the reads were mapped to our WL genome using bowtie2 (*Li and Durbin, 2009*) using default parameters. Reads were filtered for a mapping quality of 20 and BAM files were sorted by coordinates using samtools. The coverage of both W and WL libraries mapping to our WL genome was calculated using the samtools depth function. The pool-seq reads are deposited under BioProject PRJNA562690, SRA SRR10030338 for the winged reads and SRA SRR10030337 for the wingless reads.

### Biotype genomes

In addition to the reference pea aphid genome which is *api* wingless homozygous, we obtained the sequence of 22 additional pea aphid genomes: nine carrying only the winged allele and 13 the

wingless allele. These genotypes were collected in the wild as parthenogenetic females, on distinct legume species (*Guyomar et al., 2018*; *Gouin et al., 2015*). Biotypes (populations specialized on distinct legume hosts) of these genotypes were assigned based on their microsatellite profiles (*Peccoud et al., 2009b*) and representatives were then selected for genome resequencing (*Gouin et al., 2015*). Lines were sequenced to 17X to 30X coverage with Illumina 100nt paired-end reads. The reads of each sample were aligned to the pea aphid reference genome with default settings in bowtie2 (version 2.2.1) (*Langmead et al., 2009*). The consensus sequence of each sample was acquired using the recommended pipeline in samtools (version 1.3.1) and bcftools (version 1.3).

## Nanopore sequencing and genome assembly

Two F2 lines, one homozygous winged and the other homozygous wingless, were generated from the *api* heterozygous F1 (W/WL) line. The DNA of nymphs (first through fourth instars) was isolated using the Qiagen Gentra Puregene Tissue Kit according to the manufacturer's instructions. DNA was quantified on a Qubit and a Nanodrop and visualized on a pulse filed gel to verify its large size. For Nanopore sequencing, 2 µg of DNA was used to prepare a 1D sequencing library (SQK-LSK108) according to the manufacturer's instructions (1D gDNA long reads without BluePippin protocol), including the FFPE repair step. 75 µL of the library was immediately loaded onto an R9.4 flow cell prepared according to the manufacturer's instructions and run for approximately 48 hr. Basecalling was completed using ONT Albacore Sequencing Pipeline Software version v2.2.6. 14 GB of Nanopore reads were generated for the winged line and 7 GB reads were generated for the wingless line. OntCns, a customized version of MECAT (*Xiao et al., 2017*) was used to assemble the Nanopore reads of the W line. All parameters were kept the same as the example included in the package manual except for the OntCns2Consensus step, in which the coverage cutoff (-c) was set to 5. Minimap2 (*Li, 2018*) and Miniasm2 (*Li, 2016*) was used for genome assembly of the WL genome. An additional 36 GB of WL/WL Pacbio reads were combined with the 7 GB of Nanopore reads. Preset parameters (-x ava-pb) were used in the Minimap step. Assembly polishing was performed slightly differently for the two assemblies. First, 4 rounds of RACON (*Vaser et al., 2016*) (two rounds of Nanopore and two rounds of Pacbio) were performed to improve the WL assembly since there is no consensus step in the Miniasm assembler (there is a consensus step in the MECAT assembler). We aligned the Nanopore reads and Pacbio reads using Minimap2 with preset parameters (-x ava-ont and -x ava-pb) respectively. Next, 10 rounds of Pilon were used to further polish the WL assembly (*Walker et al., 2014*). We aligned 84X W and 48X WL Illumina paired-end (2 × 100 nt) reads to the assembly using bowtie2 (*Langmead and Salzberg, 2012*) and then ran Pilon on the sorted bam files using default settings. The same Pilon polishing process has been performed for the MECAT assembly. The only difference is that we used 68 × 100 nt F2(wl) and 44X F2(w) Illumina paired-end (2 × 100 nt) reads for the alignment. The wingless reads are deposited under accession SRR8306868 (wingless) and SRR10028116 (winged).

## Other genomes

We sequenced the genome of the rose aphid (*Macrosiphum rosae*) using 2 × 150 nt Illumina reads to approximately 105x coverage. Abyss 2.13 was used to assemble the genome, with kmer set to 107 and other parameters left as default. This Whole Genome Shotgun project has been deposited at GenBank under the accession BioProject: PRJNA576965. We used publicly available data for the peach-potato (*Myzus persicae*), the soybean (*Aphis glycines*), and the bird cherry-oat (*Rhopalosiphum padi*) aphid genomes, available at Aphidbase.com. We also used information from a genome produced by the 10XGENOMICS company (here renamed *Carolinaia spp.* based on its COI sequence) available at their website: https://support.10xgenomics.com/de-novo-assembly/datasets/2.0.0/aphid).

## Association and $F_{ST}$ analysis

Two mpileup files were created from the W and WL male BAM files, mapping to either the genome recently published chromosome-length genome (*Li et al., 2019*) or to the WL genome assembly we produced here using the samtools mpileup function with the -B option to disable BAQ computation. They were further processed by the mpileup2sync.jar script in PoPoolation2 v.1.201 (*Kofler et al., 2011*) to generate synchronized mpileup files (sync files), with fastq type set to sanger and minimum

quality set to 20. For association analyses, Fisher's exact tests were performed on the mapping to our WL genome using the fisher-test.pl script included in PoPoolation2 with a window-size of 1, step-size of 1, minimum coverage of 2, and minimum allele count of 2. $F_{ST}$ values were calculated on the mapping to the whole-X-chromosome assembly (*Li et al., 2019*) with the fst-sliding.pl script included in PoPoolation2 with a window size set to 20 kb, a step size set to 10 kb, minimum coverage set to 30, a maximum coverage set to 200, pool size set to 44. We filtered the results by the fraction of the window having sufficient coverage greater than 0.05.

## Species and follistatin gene phylogenies

Two *fs* genes were identified in the *A. pisum* reference genome (v3.0): XM_003246488 and XM_029488805. We found an additional copy in the insertion, which in v2.0 of the genome was referred to as ACYPI065759. They were renamed as follows: XM_003246488: *fs-1*; XM_029488805: *fs-2*; ACYPI065759: *fs-3*. The *fs* homologs in the peach-potato (*Myzus persicae*), the soybean (*Aphis glycines*), and the bird cherry-oat aphids (*Rhopalosiphum padi*) were identified by blastp using the pea aphid *fs-1*. Tblastn was used to identify the homolog in the 10X genomics, *Carolinaia spp.* genome. Finally, a blastn was used to find the *fs* gene sequence from the rose aphid. MAFFT (v.7.313) was used for alignment and Raxml (v.8.2.11) was used to build a maximum likelihood tree of the aligned coding sequences. Rapid bootstrap analysis (-f a) was used to search for the best-scoring maximum likelihood tree in one program run. The GTRGAMMA model was selected (-m GTRGAMMA) and 1000 bootstrap replicates were performed. Random seed number for the starting trees (-p 123321) and Bootstrap (-x 12321) were set accordingly.

Five housekeeping genes (Elongation factor 1α, 18S ribosomal RNA, 12S ribosomal RNA, Cytochrome b and Cytochrome c oxidase subunit I) were selected to construct the phylogeny of six aphid species: the pea aphid (*A. pisum*), the rose aphid (*Macrosiphum rosae*), a *Carolinaia spp.* aphid, the peach-potato aphid (*Myzus persicae*), the bird cherry-oat aphid (*Rhopalosiphum padi),* and the soybean aphid (*Aphis glycines)*. The sequences were concatenated together for each species and then aligned using MUSCLE (v.3.8.31). The aligned sequences were further curated using Gblock (http://www.phylogeny.fr/) to eliminate poorly aligned positions and divergent regions. Then the same settings of Raxml as in the *follistatin* phylogeny were adopted to construct the maximum likelihood tree.

## Annotations

Augustus v.3.3.1 (*Sommerfeld et al., 2009*) was used for annotations. Species parameters were set to the pea aphid, *Acyrthosiphon pisum,* since there is a pretrained model.

## SplitsTree

SplitsTree v.4.14.6 (*Huson and Bryant, 2006*) was used to generate unrooted phylogenetic networks. Two regions were analyzed for individual biotypes on the *api* scaffold: inside *api* (positions 378,979–387,730) and outside *api* (positions 1,235,176–1,245,122). All parameters were left as default.

## dS biotype calculations

The bedtools (v.2.26.0) getfasta function was used to extract CDS sequences from the consensus sequences of the Lap1 and Ms3 genomes based on the annotation file AphidBase_OGS2.1b_-withCDS.gff3, available at aphidbase.com. A modified perl script of SNAP (Synonymous Non-synonymous Analysis Program, v.2.1.1), (*Korber, 2000*) was used to calculate dS values with the Jukes-Cantor correction. Results were filtered to only include CDS lengths greater than or equal to 300 and less than or equal to 3000.

## Reverse-transcriptase quantitative PCR (RT-qPCR)

We used the two *api* homozygous F2 lines to collect male embryos and first instar nymphs from winged and wingless genotypes. We dissected stage 18 (*Miura et al., 2003*) embryos from the females or collected first instar nymphs and verified that they were male via PCR (primers 5' ATCGA TGCTTTTGAATTGTTTTAC 3' and 5' TGTAGGGTCTCTTGAAGTTGTTTG 3') followed by restriction digest. This primer set targets a heterozygous recognition sequence for Taq1alpha on the X

chromosome and males only have one X chromosome; thus, they only display one of the two possible bands shown by females. Five biological replicates were included. Each embryo replicate contained 20 embryos from 6 to 10 females, while each first instar nymph replicate contained 10 individuals produced by 3 to 5 females. RNA extraction was performed using Trizol and cDNA synthesis via the BioRAD iScript cDNA kit. Quantitative PCR was performed on a Bio-Rad CFX-96 Real-Time System using 12 µL reactions of 40 ng cDNA, 1X PCR buffer, 2 nM Mg+2, 0.2 nM dNTPs, 1X EvaGreen, and 0.025 units/µL Invitrogen Taq with the following conditions: 95°C 3 min, 40x (95°C 10 s, 55°C 30 s). Primer concentrations were optimized to $100 \pm 5\%$ reaction efficiency with an $R^2$ value of >0.99 [G3PDH (XM_001943017): 400 nM Forward primer, 350 nM Reverse primer; NADH (NM_001162323): 350 nM F, 300 nM R primer; 2281: 175 nM; 25525: 150 nM; 25532: 250 nM; and 25533: 150 nM]. Each of the five biological replicates was run on a single plate, with three technical replicates of each reaction. $\Delta$Ct values were calculated by subtracting the average $C_T$ value of the two endogenous controls (G3PDH and NADH) from the $C_T$ values of each target gene. For each pair of winged and wingless samples, $\Delta C_T$ values were analyzed using two-sided t-tests after checking for normality. The endogenous control primers used were GAPDH 5' CGGGAATTTCATTGAACGAC 3' and 5' TCCACAACACGGTTGGAGTA 3'; and NADH: 5' CGAGGAGAACATGCTCTTAGAC 3' and 5' GATAGCTTGGGCTGGACATATAG 3'. The other gene-specific primers were *frs2*: 5' AATGACACAAATGTGGCTGAAG 3' and 5' CAGGGCTTAACTCAAGGTATGG 3'; *sam50*: 5' CCTATGACATTACGTGGTTTCC 3' and 5' TTCGGTTACTTTCCATGCC 3'; *as1*: 5' GACCTGACGATTTTGAACACAG 3' and 5' TTATCTCCCATCATCAGCATTC 3'; *cdg*: 5' AAGATGCTTCTTCTTCAACACC 3' and 5' CCCTGCAGCTTGTTCTACATTA 3'.

## Reverse-transcriptase PCR (RT-PCR)

We used the *api* mapping F1 line (parental line of the two F2 lines) to collect embryonic, 1st instar, and 2nd instar males and females. For males, we verified that each individual was a male via PCR (primers 5' TGGTACATATCAGCTATCAGCACA 3' and 5' ACACAAGTTATTTCAGTTGCTTAGG 3') followed by restriction digest. This primer set targets a heterozygous recognition sequence on the X for Taq1alpha, so that males will only display one of the two possible bands depending on wing morph. We pooled groups of 20 males for RNA extraction. Wing phenotype is environmentally determined in females, so there are no genetic differences between them. To collect predominantly winged female samples, we crowded three-day-old asexual adult wingless females in groups of 12 for 24 hr. Adults were then placed on plants to lay offspring. Individuals from embryonic stage 18, the 1st nymphal instar, and the 2nd nymphal instar were collected and pooled in groups of 20 for RNA extraction. cDNA conversion was via the TransScript cDNA synthesis kit. To target *fs-3*, we used the primers 5' CTTTGCACTTTGCGCACGA 3' and 5' GCTGAACGCTTTTCGAAAGAAC 3' and a 65°C annealing temperature. For *fs-2*, we used the primers 5' GCGTCCAACGTACATATATAGAAA 3' and 5' ACACACCATTGGCATCACAT 3' and a 60°C annealing temperature. The G3PDH primers were the same as used for the RT-qPCR. 40 ng of cDNA was used for each reaction.

## Winged allele frequency estimation in pea aphid biotypes

We analyzed the *api* allele frequency of pooled samples from 12 of the biotypes (*Guyomar et al., 2018*). Reads were mapped to the reference genome (AphidBase 2.0) using bowtie2 (v2.2.9) with default settings. Samtools (v1.7) was used to generate mpileup files and popoolation2 (v2.1201) were used to generated sync files which contain the read depth of each nucleotide at each location (all default settings except minimum mapping quality (-q for samtools and —min-qual for popoolation2) was set to 20). Allele frequency was calculated based on the nucleotide frequency at the marker location (position 55599 on scaffold GL349773).

## Acknowledgements

We gratefully thank Jen Keister for technical assistance. This research was supported by award R01GM116867 from the National Institute of General Medical Sciences to JAB. Pool-seq sequencing was performed at the University of Rochester Genomics Research Center. JCS was supported by the Agence Nationale de la Recherche (grant ANR-11-BSV7-007) and France génomique (project 62

AAP 2009/2010). We thank Prof. Shin-Ichi Akimoto who provided samples and *api* phenotypes for pea aphid lines 09003A, Sap05VC7 and Iwamizawa.

## Additional information

### Funding

| Funder | Grant reference number | Author |
|---|---|---|
| National Institute of General Medical Sciences | R01GM116867 | Jennifer Brisson |
| Agence Nationale de la Recherche | ANR-11-BSV7-007 | Jean-Christophe Simon |

The funders had no role in study design, data collection and interpretation, or the decision to submit the work for publication.

### Author contributions

Binshuang Li, Data curation, Formal analysis, Validation, Investigation; Ryan D Bickel, Conceptualization, Formal analysis, Investigation, Methodology; Benjamin J Parker, Omid Saleh Ziabari, Fangzhou Liu, Investigation, Methodology; Neetha Nanoth Vellichirammal, Formal analysis, Investigation, Methodology; Jean-Christophe Simon, Resources, Writing - review and editing; David L Stern, Conceptualization, Formal analysis, Investigation, Visualization, Methodology; Jennifer A Brisson, Conceptualization, Data curation, Formal analysis, Supervision, Funding acquisition, Validation, Investigation, Visualization, Methodology, Project administration

### Author ORCIDs

Binshuang Li (ID) https://orcid.org/0000-0002-3643-747X
Benjamin J Parker (ID) https://orcid.org/0000-0002-0679-4732
David L Stern (ID) http://orcid.org/0000-0002-1847-6483
Jennifer A Brisson (ID) https://orcid.org/0000-0001-7000-0709

### Decision letter and Author response

Decision letter https://doi.org/10.7554/eLife.50608.sa1
Author response https://doi.org/10.7554/eLife.50608.sa2

## Additional files

### Supplementary files

- Source data 1. Genotypes of F2 individuals used for QTL analysis.

- Source data 2. $F_{st}$ values for winged versus wingless pool-seq comparisons.

- Source data 3. Data from RT-qPCR analysis.

- Source data 4. Alignment of different biotype individuals outside of the *api* region.

- Source data 5. Alignment of different biotype individuals inside the *api* region.

- Source data 6. Between biotype dS comparisons.

- Supplementary file 1. ORFs inside and directly outside of the *api* region.

- Supplementary file 2. Synonymous and non-synonymous substitution rates based on codon-aligned nucleotide sequences.

- Supplementary file 3. Collection locations for winged and wingless males used in the pool-seq study.

- Transparent reporting form

## Data availability

Data generated and used for this study are included in the manuscript, supporting files, or source data files. Sequencing data are deposited under accession codes SRR8306868, SRR10028116, SRR10030338, and SRR10030337. Source data files are provided for Figures 1-3.

The following datasets were generated:

| Author(s) | Year | Dataset title | Dataset URL | Database and Identifier |
|---|---|---|---|---|
| Binshuang Li, Ryan D Bickel, Benjamin J Parker, Omid Saleh Ziabari, Fangzhou Liu, Neetha Nanoth Vellichirammal, Jean-Christophe Simon, David L Stern, Jennifer A Brisson | 2019 | Nanopore sequencing of an Acyrthosiphon pisum line | https://www.ncbi.nlm.nih.gov/sra/SRR8306868 | NCBI Sequence Read Archive, SRR8306868 |
| Binshuang Li, Ryan D Bickel, Benjamin J Parker, Omid Saleh Ziabari, Fangzhou Liu, Neetha Nanoth Vellichirammal, Jean-Christophe Simon, David L Stern, Jennifer A Brisson | 2020 | Nanopore sequencing of an Acyrthosiphon pisum line (aphicarus W/W) | https://www.ncbi.nlm.nih.gov/sra/SRR10028116 | NCBI Sequence Read Archive, SRR100 28116 |
| Binshuang Li, Ryan D Bickel, Benjamin J Parker, Omid Saleh Ziabari, Fangzhou Liu, Neetha Nanoth Vellichirammal, Jean-Christophe Simon, David L Stern, Jennifer A Brisson | 2020 | Paired end sequencing of pooled Acyrthosiphon wingless males | https://www.ncbi.nlm.nih.gov/sra/SRR10030337 | NCBI Sequence Read Archive, SRR10030 337 |
| Binshuang Li, Ryan D Bickel, Benjamin J Parker, Omid Saleh Ziabari, Fangzhou Liu, Neetha Nanoth Vellichirammal, Jean-Christophe Simon, David L Stern, Jennifer A Brisson | 2020 | Paired end sequencing of pooled Acyrthosiphon winged males | https://www.ncbi.nlm.nih.gov/sra/SRR10030338 | NCBI Sequence Read Archive, SRR10030 338 |
| Binshuang Li, Ryan D Bickel, Benjamin J Parker, Omid Saleh Ziabari, Fangzhou Liu, Neetha Nanoth Vellichirammal, Jean-Christophe Simon, David L Stern, Jennifer A Brisson | 2020 | Rose aphid genome | https://www.ncbi.nlm.nih.gov/bioproject/?term=PRJNA576965 | NCBI BioProject, PRJNA576965 |

The following previously published dataset was used:

| Author(s) | Year | Dataset title | Dataset URL | Database and Identifier |
|---|---|---|---|---|
| Gouin A, Legeai F, Nouhaud P, Whibley A, Simon JC, Lemaitre C | 2015 | Whole-genome re-sequencing of non-model organisms: lessons from unmapped reads | https://www.ncbi.nlm.nih.gov/bioproject/?term=PRJNA255937 | NCBI BioProject, PRJNA255937 |

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
