## [Decision Letter]

**Acceptance summary:**

Adaptive morphological traits have been mapped primarily to cis-regulatory elements or coding sequences. This paper offers compelling evidence that a classic winged/wingless polymorphism in male pea aphids maps instead to a large, X-linked insertion that duplicated over 10 million years ago. The discovery of a single strong candidate gene carried on the insertion highlights the importance of radical structural rearrangements in adaptive evolution.

**Decision letter after peer review:**

Thank you for submitting your article "A large genomic insertion containing a duplicated *follistatin* gene is linked to the pea aphid male wing dimorphism" for consideration by *eLife*. Your article has been reviewed by three peer reviewers, one of whom is a member of our Board of Reviewing Editors, and the evaluation has been overseen by Patricia Wittkopp as the Senior Editor. The following individuals involved in review of your submission have agreed to reveal their identity: Nancy Moran (Reviewer #2); Leif Andersson (Reviewer #3).

The reviewers have discussed the reviews with one another and the Reviewing Editor has drafted this decision to help you prepare a revised submission. Please note, however, that the requested changes are significant and the suitability for *eLife* following review of a revised version is thus more uncertain than usual. *eLife*

Summary:

Li et al. report their comprehensive effort to identify the genetic basis of a wing presence/absence polymorphism in the pea aphid. The authors introduce the topic well, highlighting the pervasiveness of wing loss over insect evolution and the power of within species polymorphism to map the genetic basis of wing loss. Unlike many genotype-phenotype studies published over the past decade that map to a cis-regulatory element or a nonsynonymous site, the authors report a structural variant that determines winglessness. Importantly, single- molecule sequencing was required to make this discovery, highlighting the manuscript's timeliness.

The authors validate and narrow a previously reported X-linked region controlling the wing polymorphism. They use three different methods – QTL on an F2 population, Fst analysis of US-wide populations of the pea aphid, and then an association study on so-called biotypes that are deeply diverged lineages within the species complex. Assembly of long-reads revealed a 120kb insertion only in the wingless morph. The authors go on to show that the insertion onto the X from an autosomal region is quite old. Using an additional aphid species genomes, including a previously unsequenced species, the authors infer that the insertion is unique to the pea aphid, at least among this set of species. To define a single candidate gene, the authors use several approaches to narrow the 12 genes encoded on the insertion plus those flanking the insertion. The authors identify a *follistatin* gene duplicate as the likely candidate and provide some evidence from *Drosophila* that this gene may in fact mediate wing development.

All three reviewers found the broad question compelling and the mapping to a structural variant convincing and significant. The three reviewers also determined that additional data was needed to support *follistatin-3* as the likely causal locus. Suggested experiments and additional analyses below would offer either more robust support for *follistatin* or instead support of another gene encoded on the insertion or possibly a gene in linkage with the insertion. Additionally, interpretation of the *follistatin* gene tree warrants more caution and alternative methods are required to detect purifying/negative selection on the candidate causal locus. Please find specific questions/suggestions related to these two points as well as other essential comments below:

Essential revisions:

1) The authors failed to evaluate sufficiently the 11 genes encoded on the insertion and *as1* outside the insertion. Addressing the questions below with additional analyses and/or experiments is necessary to identify a convincing candidate gene.

a) Do these 11 gene (in males) as well as *fs-2* (in males and females) show a developmental pattern of expression different from *fs-3*, consistent with *fs-3* winglessness-specificity? Additional RT-PCR (or better, RT-qPCR) would address these important questions. Note that the current RT-PCR data are missing a loading control and one reviewer noted the absence of a genomic DNA contamination control.

b) Do these 11 genes evolve at a faster rate than *fs-3*, consistent with lack of functional constraint? Are the open reading frames intact? Molecular evolution analyses would address this point. Given the ancient origins of the insertion. there has been ample time to accumulate mutations at non-functional genes.

c) The *as1* expression data deserve more discussion – the differential expression data are actually compelling.

d) A more convincing approach to support *fs-3* as the causal locus would be RNA FISH on the relevant developing tissue, though is not required if additional supporting data for *follistatin-3* emerge from the above experiments.

e) Please acknowledge explicitly that a transgene rescue experiment is ultimately required to demonstrate causality.

2) The *fs* paralog tree interpretation is problematic. The support value for the *fs-2/fs-3* clade is extremely low, suggesting that the topology should collapse into a polytomy. The ambiguity undermines the inference that an ancestral *fs-2* was lost. Please clarify. Moreover, if the authors find additional support for the presented topology, could it be instead that the ancient *fs-3* duplication represented an origin of male wing polymorphism, and that *Carolinaia* and *M. rosae* lost wingless males, as did some of the *A. pisum* biotypes? Please address this alternative interpretation.

3) I am not familiar with methods that allow inference of purifying selection from paralog comparisons (rather than ortholog comparisons). I don't have a great suggestion here for inferring purifying selection on a young gene duplicate (except, possibly, comparing to other genes in the insertion that may be evolving under no constraint and so have degenerated-see 1b).

4) There are weaknesses with Figure 1—figure supplement 3 and the argument that wingless males are rare in general in related aphid species (and so the wingless gene and phenotype is derived in *A. pisum*). I agree that this example of the wingless state is most likely derived, resulting from this insertion. However, the insertion may not be unique to *A. pisum* alone given the apparent age of the event. Importantly, the phylogeny in Figure 1—figure supplement 3 (from Hardy et al.) is not well-supported. It was based on an alignment of 4800 nucleotide sites but most of the taxa have missing data and no confidence metrics were included (What are units in the scale bar of Figure 1—figure supplement 3? What is '20.0')? I suggest summarizing winged/wingless males on the well-established Macrosiphini tree.

5) The last paragraph of the Introduction says that the majority of species in this group have winged males. Possibly the proportion is indeed more than half, but if you look in Blackman and Eastop (Aphids on the world's plants), one or more species in almost all of the genera of Figure 1—figure supplement 3 are known to have wingless males. By chance it seems there are fewer in the set represented in the Figure 1—figure supplement 3 tree – numerous of these are host plant-alternating species and therefore must have winged males. (Of course, if other species descend from host alternating species, then winged males have to be ancestral…). The idea that wingless males are derived seems almost certainly correct, but this point needs to be clarified in the context of the species presented to establish this point.

[Editors' note: further revisions were suggested prior to acceptance, as described below.]

Thank you for resubmitting your work entitled "A large genomic insertion containing a duplicated *follistatin* gene is linked to the pea aphid male wing dimorphism" for further consideration by *eLife*. Your revised article has been evaluated by Patricia Wittkopp as the Senior Editor and a Reviewing Editor.

The manuscript has been improved but there are some remaining issues that need to be addressed before acceptance, as outlined below:

1) The Supplementary file 1, which includes the annotation of the predicted ORFs both inside and outside the insertion, does not sufficiently address the reviewers' concerns (#1). There are no data in this file supporting the statement that all "g" ORFs represent repetitive sequences in the genome. If indeed all ORFs occur many times, additional data demonstrating this point should appear in this file or instead as a distinct supplementary file/figure. For those ORFs that are unique (if any), data on their expression in wingless males is important for supporting the ultimate focus on *fs-3*. It appears that the ORFs have predicted splice junctions, which could be used to support or reject the presence of a transcript. Addressing this concern is imperative given the inability to conduct knockout or transgene experiments.

2) The new RT-PCR data are certainly promising but the quality of the gel image is poor. It is difficult to discern that "…wingless males 266 expressed *fs-3* increasingly across development." Either additional product should be run to make clearer the result or, as suggested in the first Decision Letter, RT-qPCR should be performed (particularly given the use of the term "increasingly"). It is surprising that the authors chose to run RT-qPCR for some of the data (i.e., the genes outside the insertion) but RT-PCR for the more important focal gene, *fs-3*.

3) The rejection of *as1* may be warranted based on statistical criteria but given the RT-qPCR data and that it has a gene name, it would be helpful to at least know its function to further support the decision to reject it as a candidate gene.

---

## [Author Response]

Essential revisions:1) The authors failed to evaluate sufficiently the 11 genes encoded on the insertion and as1 outside the insertion. Addressing the questions below with additional analyses and/or experiments is necessary to identify a convincing candidate gene.a) Do these 11 gene (in males) as well as fs-2 (in males and females) show a developmental pattern of expression different from fs-3, consistent with fs-3 winglessness-specificity? Additional RT-PCR (or better, RT-qPCR) would address these important questions. Note that the current RT-PCR data are missing a loading control and one reviewer noted the absence of a genomic DNA contamination control.

We have collected new cDNA from males and females to perform the *fs-2* and *fs-3* RT-PCRs. We collected a different range of stages this time: instead of collecting all stages of development, we focused specifically on embryos, 1^st^ instar nymphs, and 2^nd^ instar nymphs. Our rationale for this choice was twofold. First, these are likely the most relevant stages of expression, as noted in the paper. Second, this makes the results more consistent with the RTqPCR presented in Figure 2—figure supplement 2.

The results are presented in the updated Figure 2. Recall that our purpose for the RT-PCR of *fs-2* was to show that it is expressed. Figure 2D shows that wingless males exhibit expression of *fs-3* and winged males do not (because they do not have the insertion and thus do not have the *fs-3* gene). Figure 2D also shows a positive control (G3PDH) for these same samples. This positive control also works as a negative control for genomic DNA contamination of the cDNA, since it has an amplicon size difference between genomic DNA and cDNA.

The *fs-2* gene also appears to be expressed in some samples (please Figure 2D), though not in the same pattern at *fs-3*.

We hope this addresses the reviewers’ concerns about the *fs* genes. As for the other 11 genes, please see the next response.

b) Do these 11 genes evolve at a faster rate than fs-3, consistent with lack of functional constraint? Are the open reading frames intact? Molecular evolution analyses would address this point. Given the ancient origins of the insertion. there has been ample time to accumulate mutations at non-functional genes.

We understand from these critiques that we did not provide enough information about these 11 genes. In retrospect it was probably inappropriate to describe them as genes. These are predicted ORFs using automated annotation. All 11 ORFs have strong similarity to sequences that are present many times in the pea aphid genome. Thus, the 11 ORFs are highly repetitive sequences and thus are likely non-functional and unlikely to play a role in the *api* phenotype. We have now revised the paragraph describing the 11 genes:

“Annotation software (Augustus v.3.3.1) predicted 12 open reading frames (ORFs) in the wingless *api* insertion (Figure 2A), only one of which has homologous sequence in the insertion’s autosomal source location. The other 11 ORFs are each composed of sequences that are highly repetitive across the genome, with many apparently derived from transposable elements (Supplementary file 1).”

Also as noted in this added text, none of these ORFs have homologous copies present in the original, autosomal location. Therefore, we cannot do any molecular evolution analyses among paralogs as we do with the *follistatin* genes.

c) The as1 expression data deserve more discussion – the differential expression data are actually compelling.

After the appropriate multiple comparison correction the p-value is 0.10, so given that we tested multiple genes, this difference does not meet statistical significance, is likely due to chance, and is not compelling evidence. We pointed out in the text that the uncorrected p-value was 0.013, and then followed that by stating “We therefore found no strong evidence that these genes contribute to the male wing dimorphism.” We are therefore trying to be as transparent as possible about this result.

d) A more convincing approach to support fs-3 as the causal locus would be RNA FISH on the relevant developing tissue, though is not required if additional supporting data for follistatin-3 emerge from the above experiments.

We agree and hope to eventually do this experiment. Unfortunately, there are many candidate stages and tissues and this experiment is outside the scope of this paper. We feel the qPCR supports our conclusions.

e) Please acknowledge explicitly that a transgene rescue experiment is ultimately required to demonstrate causality.

Done. We added this sentence:

“A transgene rescue or mutational knock-out experiment (ideally both) are ultimately required to determine whether *fs-3* causes the male wing polymorphism.”

2) The fs paralog tree interpretation is problematic. The support value for the fs-2/fs-3 clade is extremely low, suggesting that the topology should collapse into a polytomy. The ambiguity undermines the inference that an ancestral fs-2 was lost. Please clarify. Moreover, if the authors find additional support for the presented topology, could it be instead that the ancient fs-3 duplication represented an origin of male wing polymorphism, and that Carolinaia and M. rosae lost wingless males, as did some of the A. pisum biotypes? Please address this alternative interpretation.

Yes, it is not well supported. We have now collapsed the topology, as suggested. We collapsed two branches and now the presentation is one of a polytomy. The relevant portion of the paragraph, now reads:

“The most parsimonious explanation for these patterns is that the initial duplication that gave rise to the *fs-2/3* lineage arose after the split of the pea aphid and the peach-potato aphid, and this duplicate copy was subsequently lost in the rose aphid lineage. […] Future sequencing of more aphid genomes would provide insight into the timing of the emergence of these paralogs.”

3) I am not familiar with methods that allow inference of purifying selection from paralog comparisons (rather than ortholog comparisons). I don't have a great suggestion here for inferring purifying selection on a young gene duplicate (except, possibly, comparing to other genes in the insertion that may be evolving under no constraint and so have degenerated-see 1b).

As far as we know, this is the best tool we have available. Whole genome duplication studies frequently use dN/dS of paralogs for similar reasons (e.g., Berthelot et al. 2014 Nat Comm paper on trout genome whole-genome duplication). We would be happy to do further work to address this if an appropriate analysis can be recommended.

4) There are weaknesses with Figure 1—figure supplement 3 and the argument that wingless males are rare in general in related aphid species (and so the wingless gene and phenotype is derived in A. pisum). I agree that this example of the wingless state is most likely derived, resulting from this insertion. However, the insertion may not be unique to A. pisum alone given the apparent age of the event. Importantly, the phylogeny in Figure 1—figure supplement 3 (from Hardy et al) is not well-supported. It was based on an alignment of 4800 nucleotide sites but most of the taxa have missing data and no confidence metrics were included (What are units in the scale bar of Figure 1—figure supplement 3? What is '20.0')? I suggest summarizing winged/wingless males on the well-established Macrosiphini tree.

Please see combined answer for 4 and 5 under 5 below.

5) The last paragraph of the Introduction says that the majority of species in this group have winged males. Possibly the proportion is indeed more than half, but if you look in Blackman and Eastop (Aphids on the world's plants), one or more species in almost all of the genera of Figure 1—figure supplement 3 are known to have wingless males. By chance it seems there are fewer in the set represented in the Figure 1—figure supplement 3 tree – numerous of these are host plant-alternating species and therefore must have winged males. (Of course, if other species descend from host alternating species, then winged males have to be ancestral…). The idea that wingless males are derived seems almost certainly correct, but this point needs to be clarified in the context of the species presented to establish this point.

We have now exchanged the tree in Figure 1—figure supplement 3 for the tree from a phylogenetic study of the Macrosiphini, specifically – from Choi et al. 2018 (Molecular phylogeny of Macrosiphini (Hemiptera: Aphididae): An evolutionary hypothesis for the Pterocomma-group habitat adaptation). And we agree that the insertion is likely not unique to the pea aphid given the likely age of the insertion event. We see that the text, as it was written, implied this, even though we never meant to imply this. We have thus added these sentences:

“Given the age of this insertion, the region is likely not unique to pea aphids, but rather could be responsible for the wingless male phenotype observed in other, related species as well. […] It will be interesting to interrogate other, closely related wingless and dimorphic species to determine whether or not the insertion is present in them and thus responsible for the wingless phenotype in other species.”

To the broader point – we think there have been many transitions between winged and wingless males across the aphid phylogeny. This is an idea we are actively pursuing with further work. For now, we have revised the text to say: “The majority of the species closely related to the pea aphid produce only winged males (Figure 1—figure supplement 3), so the wingless phenotype is likely the derived phenotype.”

[Editors' note: further revisions were suggested prior to acceptance, as described below.]

The manuscript has been improved but there are some remaining issues that need to be addressed before acceptance, as outlined below:1) The Supplementary file 1, which includes the annotation of the predicted ORFs both inside and outside the insertion, does not sufficiently address the reviewers' concerns (#1). There are no data in this file supporting the statement that all "g" ORFs represent repetitive sequences in the genome. If indeed all ORFs occur many times, additional data demonstrating this point should appear in this file or instead as a distinct supplementary file/figure. For those ORFs that are unique (if any), data on their expression in wingless males is important for supporting the ultimate focus on fs-3. It appears that the ORFs have predicted splice junctions, which could be used to support or reject the presence of a transcript. Addressing this concern is imperative given the inability to conduct knockout or transgene experiments.

We have made a series of changes that we hope will help address this issue.

– As part of the pea aphid genome release in 2010, a database of pea aphid transposable element consensus sequences was created. We have used blastn to query this database and have reported the results in this supplementary table. All but two of the ORFs from the insertion hit this repeat database. One of the ORFs is our focal gene, the *follistatin* copy (*fs-3*). The other is g6. This ORF has a conserved DDE_Tnp4 domain, part of the DDE superfamily of endonucleases that are likely transposases. Although a blastn of the nucleotide sequence does not hit the pea aphid TE database, a tblastn using the g6 predicted protein sequence reveals that similar sequences are present over 100 times in the pea aphid genome. This is consistent with g6 being a low copy number variant of an extremely common superfamily of TEs present in the pea aphid genome. We have updated the text to say: “Annotation software (Augustus v.3.3.1) predicted 12 open reading frames (ORFs) in the wingless *api* insertion. […] The remaining ORF, *fs-3*, is not repetitive and, importantly, is the only ORF that has homologous sequence in the insertion’s autosomal source location.”

– We have changed the descriptions of these genes in the supplementary file. Rather than using an aggregation of blast hits, we have made the information more systematic. Now the annotations are restricted to conserved domains and their evalues. We have also added a column of information that summarized the other columns.

– We have reduced the number of genes listed in the supplementary file. Specifically, we have deleted two of the gene models (g12 and g14) that were outside of the insertion. Our previous Figure 2A showed annotations of genes from the genome annotation version 2.1b. A new annotation came out later last year and these two genes are *not* in that annotation. Our edits make the manuscript more consistent because we are using v3.0 for the genomic sequence elsewhere in the manuscript. We have also removed reference to these genes in the text. We have also changed out all ACYPI IDs (version 2.1b reference IDs) in the text to the new, v3.0 names of genes except in Supplementary file 1 where we have listed both names.

– In case it was previously unclear, the gene models within the insertion are our own annotations, not genome v3.0 annotations. This is because much of the insertion sequence is not found in this version of the genome. We could, therefore, not use it. We have now stated this clearly in the text and in the legend for Figure 2.

Our intention with these changes was to make the data on all the ORFs/genes more systematic and to reflect the latest official build and annotation of the genome as much as possible.

2) The new RT-PCR data are certainly promising but the quality of the gel image is poor. It is difficult to discern that "…wingless males 266 expressed fs-3 increasingly across development." Either additional product should be run to make clearer the result or, as suggested in the first Decision Letter, RT-qPCR should be performed (particularly given the use of the term "increasingly"). It is surprising that the authors chose to run RT-qPCR for some of the data (i.e., the genes outside the insertion) but RT-PCR for the more important focal gene, fs-3.

We have re-run the PCRs to present better gel images. And, importantly, we have removed reference to any sort of relative expression in the text because we should not have used terms like “increasingly”.

We ran RT-qPCR for the genes outside of the insertion because we thought it would be relevant to know the expression in the winged relative to the wingless morph. Our rationale was that a difference between the morphs could be driven by a regulatory change at one of these genes. We therefore presented RT-qPCR data in Figure 2—figure supplement 2. For the focal gene (*fs-3*), which is inside the insertion, this comparison is not possible; the gene is not present in the winged morph, so we cannot look at relative expression in the winged relative to the wingless morph. Any RT-qPCR, and thus any relative expression, would only be comparing expression across developmental stages. This type of RT-qPCR would be useful for beginning to unravel the functioning of this gene, but that is not the subject of this manuscript. Instead, the purpose here is to establish that this gene is expressed in the wingless males. RT-PCR establishes that it is expressed. We have added text in the manuscript about this rationale. We hope that this clearly addresses why we have performed RT-qPCR for some genes and RT-PCR for the other.

3) The rejection of as1 may be warranted based on statistical criteria but given the RT-qPCR data and that it has a gene name, it would be helpful to at least know its function to further support the decision to reject it as a candidate gene.

It would be great to know the function, but it is unknown. There are no functional domains, and all blast hits are to other aphid genes. We have now made it clear in the text that *we* named this gene *as1*. We have also now stated in the text that this gene has no known function and no conserved domains.